# Musculoskeletal Chronic Graft versus Host Disease—A Rare Complication to Allogeneic Hematopoietic Stem Cell Transplant: A Case-Based Report and Review of the Literature

Alexander Dåtland Kvinge [1,†], Tobias Kvammen [1,†], Hrvoje Miletic [2,3], Laurence Albert Bindoff [4,5] and Håkon Reikvam [1,6,*]

1   Institute of Clinical Science, Faculty of Medicine, University of Bergen, N-5021 Bergen, Norway
2   Department of Pathology, Haukeland University Hospital, N-5021 Bergen, Norway
3   Department of Biomedicine, University of Bergen, N-5009 Bergen, Norway
4   Department of Neurology, Haukeland University Hospital, N-5021 Bergen, Norway
5   Institute of Medical Science, Faculty of Medicine, University of Bergen, N-5021 Bergen, Norway
6   Department of Medicine, Haukeland University Hospital, N-5021 Bergen, Norway
*   Correspondence: hakon.reikvam@uib.no; Tel.: +55-97-5000; Fax: +55-97-2950
†   These authors contributed equally to this work.

**Abstract:** Musculoskeletal graft versus host disease (GVHD) is a rare manifestation of chronic GVHD (cGVHD) following allogeneic hematopoietic stem cell transplantation (allo-HSCT). Left untreated, the disease can cause extensive damage to muscle tissue and joints. We describe a 62-year-old male with musculoskeletal GVHD and generalized muscle pain and stiffness. In addition, we performed a systemic literature review based on published cases of musculoskeletal GVHD between 1983 and 2019. We identified 85 cases, 62% male and 38% female with an age of 4–69 years and median age of 39 years at diagnosis. The majority of patients (72%) also had manifestations of cGVHD in at least one other organ system, most frequently the skin (52%), followed by oropharyngeal mucosa (37%), and pulmonary and gastrointestinal tract (GI tract) (21%). We conclude that, while musculoskeletal cGVHD is a rare complication of allo-HSCT, it remains a serious and debilitating risk that must be considered in patients with muscle pain, muscle weakness, joint stiffness, and tissue inflammation. Early intervention is critical for the patient's prognosis.

**Keywords:** allogenic hematopoietic stem cell transplantation; graft-versus-host disease; musculoskeletal pathology; fasciitis; myositis; creatine kinase; immunosuppressive treatment

## 1. Introduction

Chronic graft versus host disease (cGVHD) is an immune-mediated condition and the most common complication of allogeneic hematopoietic stem cell transplantation (allo-HSCT). GVHD develops when immunocompetent T cells from the donor recognize antigens, known as human leukocyte antigen (HLA), from the recipient as foreign [1]. In contrast to acute GVHD (aGVHD), which mainly affects skin, liver, and gastrointestinal tract (GI tract), cGVHD can affect almost any organ system. However, the most frequent is GI tract, oropharynx, skin, eyes, urogenital tractus, lungs, and the lympho-hematopoietic system. A rare manifestation of cGVHD is musculoskeletal GVHD, potentially leading to disability, impairment, severely reduced quality of life, and may ultimately be potentially life-threatening [2,3]. The manifestations of musculoskeletal GVHD include myositis, fasciitis, contractures, and reduced range of motion in the affected joints. Pathophysiological, there is an increase in alloimmune donor T cells in the soft tissues and muscle surrounding the affected joints. Upon activation of T cells, they release cytokines, which trigger fibroblast activation and proliferation. Further, fibrosis occurs due to crosstalk between macrophages and T cells. Additionally, B-cell activation caused by the alloimmune donor T cells results

in the production of autoantibodies directed towards muscle tissue and joints [4]. Herein, we present a patient who developed general muscle pain, stiffness, and elevated levels of creatinine kinase (CK) after allo-HSCT. We discuss the clinical findings, diagnostics, and treatment options based on a systemic review of the published literature.

## 2. Case Report

A 62-year-old man underwent allo-HSCT with a reduced-intensity conditioning regime (RIC) including fludarabine and treosulfan for an unclassifiable myelodysplastic/myeloproliferative neoplasm (MDS/MPN-U). The donor was matched with a sibling sharing both haplotypes, i.e., 12/12 human leukocyte antigen (HLA) match. The bone marrow reconstituted within day +20, and at day +28, he had complete donor chimerism, which was maintained at future evaluations. At day +60 after transplant, he developed aGVHD manifestations in both the skin and GI tract, classified as grade I and grad IV, respectively, both verified by biopsies. The GI tract aGVHD was severe and needed treatment with high-dose corticosteroids, cyclosporine A, and alfa-1-antrypsine. However, the symptoms resolved; the cyclosporine A and alfa-1-antitrypsine treatment was terminated and corticosteroids slowly tapered. During gradual tapering of the prednisolone, an increase in liver enzymes was discovered, and a liver biopsy confirmed the histopathological findings of liver cGVHD that was again treated with corticosteroids in tapering doses. At the routine control 2.5 years post-transplant, the patient complained of severe generalized muscle pain and stiffness, most pronounced in the proximal part of both upper and lower extremities. He had moderate pain in the proximal extremities' muscles, most pronounced in the morning. An investigation revealed moderately elevated CK and C-reactive protein (CRP) (Table 1). Differential diagnostic considerations included viral infections, other autoimmune myositis, drug-induced side effects, or muscle cGVHD. For confirmatory diagnostics, a muscle biopsy was performed from the quadriceps, and histology revealed massive inflammation and caliber variations. A large portion of the muscle tissue was replaced by inflammatory cells, and the remaining muscle fibers were highly atrophic. Immunohistochemistry (IHC) revealed that the inflammatory cells comprised CD45+ leukocytes, along with CD4+ and CD8+ lymphocytes. Additionally, there were large numbers of CD68+ macrophages (Figure 1). These findings were consistent with a severe myositis.

**Table 1.** Diagnostic blood tests from the patient, both before and after treatment.

| Analysis | Before Treatment | Six Months after Treatment | Reference Area |
|---|---|---|---|
| Hemoglobin (g/dL) | 14.0 | 13.6 | 13–18 |
| WBC ($\times10^9$/L) | 15.3 | 7.5 | 3.5–10.0 |
| Neutrophils ($\times10^9$/L) | 11.9 | 2.6 | 1.5–7.3 |
| Lymphocytes ($\times10^9$/L) | 2.4 | 4 | 1.1–3.3 |
| Thrombocytes ($\times10^9$/L) | 291 | 250 | 150–400 |
| Ferritin (ng/mL) | 658 | 786 | 7–75 |
| CRP (mg/L) | 52 | <1 | <5 |
| Sodium (mmol/L) | 138 | 141 | 136–146 |
| Potassium (mmol/L) | 4.7 | 4.1 | 3.5–5.0 |
| ALAT (U/L) | 107 | 40 | <70 |
| ALP (U/L) | 155 | 39 | 35–105 |
| GT (U/L) | 425 | 91 | <115 |
| Bilirubin (μmol/dL) | 9 | 5 | <26 |
| LDH (U/L) | 268 | 197 | 300–600 |
| Albumin (g/L) | 38 | 42 | 36–47 |
| CK (U/L) | 1291 | 173 | <280 |

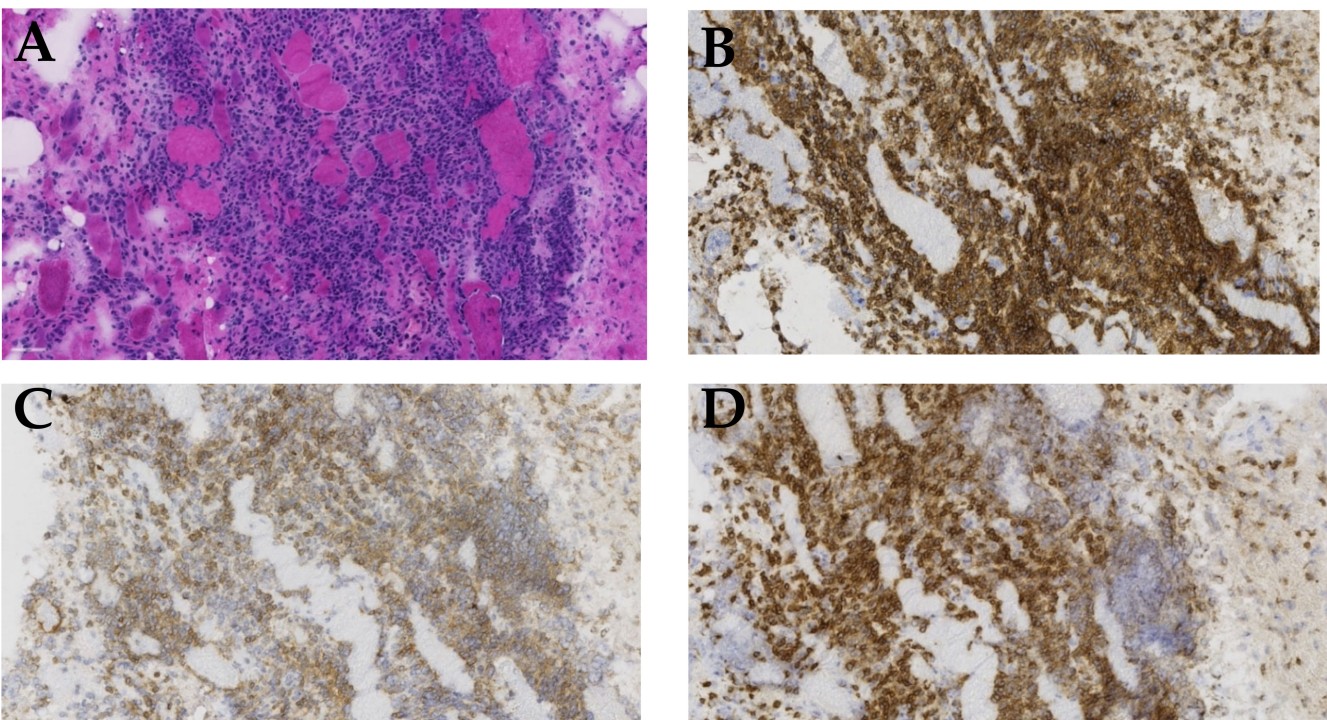

**Figure 1.** Histopathological features of muscle biopsy. This figure demonstrates the results of a biopsy from the left quadriceps femoris muscle, taken from our patients. (**A**) Hematoxylin–eosin staining reveals that large amounts of the muscle tissue have been replaced by inflammatory cells. (**B**) Immunohistochemical staining for CD45 shows massive amounts of leukocytes present in the tissue sample. (**C**) Immunohistochemical staining for CD4 shows a massive infiltration of CD4+ lymphocytes. (**D**) Immunohistochemical staining for CD8 demonstrates an extensive infiltration of CD8+ lymphocytes. All the histopathological images are shown in 40× magnification.

Based on the biochemical, histopathological, and clinical evaluations, we concluded that the patient was suffering from musculoskeletal manifestations of cGVHD. We did not perform a magnetic resonance imaging (MRI) scan, which might have shown us which muscles were affected; however, the clinical symptoms were mainly from proximal extremity muscles. Nevertheless, it is highly likely that many muscles were affected but to varying degrees. This fitted with an inflammatory condition that most often affects the proximal muscles mostly, but often patchily, as has been reported in cGVHD. Electromyography (EMG) was not carried out.

While corticosteroids have generally been considered as the first-line treatment for musculoskeletal cGVHD [2,5], our patient suffered troublesome side effects when treated with prednisolone for liver GVHD; therefore, treatment with the Janus kinase (JAK) inhibitor ruxolitinib at doses of 10 mg twice daily was initiated. This treatment showed a rapid regress of pain and muscle stiffness and normalization of the CRP and CK levels. Additionally, there was a prompt and lasting improvement of his liver enzymes (Table 1), and the current follow-up shows no signs of disease progression, full recovery of the musculoskeletal symptoms, and normalization of the CK values (Table 1).

This table shows blood work done before and after the treatment of our patient. Values before treatment are when first discovering the musculoskeletal symptoms. Values after treatment are six months after being treated with ruxolitinib. This table demonstrates different analyses performed, and it shows the obtained value, as well as the given reference area. Abbreviations: WBC, white blood cell count; CRP, C-reactive protein; ALAT, alanine aminotransferase; ALP, alkaline phosphatase; GT, gamma-glutamyl transferase; LDH, lactic dehydrogenase; and CK, creatinine kinase.

## 3. Methods

### 3.1. Strategy of Search and Data Sources

We performed a systematic search of the PubMed database using variations of the keywords "musculoskeletal", "graft versus host disease", and "GVHD", as well as "myositis" and "fasciitis". Due to the rarity of the diagnosis, no search filters or time restrictions were applied. Based on this search, we found a total of 842 articles, all of which were used in the primary screening (Figure S1).

### 3.2. Selection Criteria

During the initial screening, nonrelevant articles were excluded based on title or abstract, as were articles and cases not written in English. This left a total of 82 articles (Figure S1). In the secondary screening, articles with insufficient data and image reports were excluded and any duplicates also removed. From the remaining 30 articles, we identified 85 patients diagnosed with musculoskeletal cGVHD, who were then used in this review.

### 3.3. Data Extraction

From the 85 patients, data regarding sociodemographic status, clinical data, transplantation features, treatment, and outcomes were extracted. All data was double-checked by both primary authors.

## 4. Results

### 4.1. Etiology

The 85 cases identified included the present case, and the demographical and clinical data are presented in Table 2. The overall age ranged from 4 years to 69 years old: in women, this was 8–67 years, while, in men, it was 4–69 years. The median age of all patients was 39 years. The gender distribution was uneven, with 53 (62%) male and 32 (38%) female patients. The distribution of age and gender is presented in Figure 2.

The difference in hematological diseases varied greatly. The most common disease was acute myelogenous leukemia (AML) (39%), followed by chronic myelogenous leukemia (CML) (18%) and acute lymphoblastic leukemia (ALL) (13%). The distribution of hematological diseases is shown in Table 3.

**Table 2.** Overview of the literature search.

| Case Nr. | Sex | Age | Hem Dis | AI/Au | Diagnostic | | | Lab | Symptoms | | | | | | Treatment | | | | | | | | | Other Organs | | | | | | | | | | Treat Res | | | | |
|---|---|---|---|---|---|---|---|---|---|---|---|---|---|---|---|---|---|---|---|---|---|---|---|---|---|---|---|---|---|---|---|---|---|---|---|---|---|---|
| | | | | | Bx | Radio | EMG | CK | Weak | Pain/Joint | Swel | Fever | RoM | GC | CyA | Tacro | IG | Other | MM | Az | MA | Skin | Oral | Eyes | GI | Pulm | Liver | CNS | Joints | Heart | Res | IwD | NR | NCT | Ref. |
| 1 | F | 33 | AML | Allo | | | | <25 | | | | | | | | | | | | | | | | | | | | | | | NA | | | | [6] |
| 2 | F | 46 | NHL | Allo | | | | <25 | | | | | | | | | | | | | | | | | | | | | | | NA | | | | [6] |
| 3 | F | 55 | AA | Allo | | | | 727 | | | | | | | | | | | | | | | | | | | | | | | NA | | | | [6] |
| 4 | F | 64 | NHL | Allo | | | | 35 | | | | | | | | | | | | | | | | | | | | | | | NA | | | | [6] |
| 5 | M | 57 | AML | Allo | | | | <25 | | | | | | | | | | | | | | | | | | | | | | | NA | | | | [6] |
| 6 | F | 65 | NHL | Auto | | | | 33 | | | | | | | | | | | | | | | | | | | | | | | NA | | | | [6] |
| 7 | M | 58 | MDS | Allo | | | | 5200 | | | | | | | | | | | | | | | | | | | | | | | NA | | | | [6] |
| 8 | M | 66 | CLL | Allo | | | | 278 | | | | | | | | | | | | | | | | | | | | | | | NA | | | | [6] |
| 9 | M | 61 | AML | Allo | | | | 20 | | | | | | | | | | | | | | | | | | | | | | | NA | | | | [6] |
| 10 | M | 32 | AML | Allo | | | | 123 | | | | | | | | | | | | | | | | | | | | | | | NA | | | | [6] |
| 11 | M | 58 | MDS | Allo | | | | 84 | | | | | | | | | | | | | | | | | | | | | | | NA | | | | [6] |
| 12 | M | 65 | AML | Allo | | | | 2422 | | | | | | | | | | | | | | | | | | | | | | | NA | | | | [6] |
| 13 | M | 69 | NHL | Auto | | | | NA | | NA | | | | | | | | | | | | | | | | | | | | | NA | | | | [6] |
| 14 | M | 52 | CLL | Allo | | | | 98 | | | | | | | | | | | | | | | | | | | | | | | NA | | | | [6] |
| 15 | M | 34 | CML | Allo | | | | 1027 | | | | | | | | | | | | | | | | | | | | | | | | | | | [3] |
| 16 | F | 46 | AML | Allo | | | | NA | | | | | | | | | | | | | | | | | | | | | | | | | | | [4] |
| 17 | F | 36 | MM | Allo | | | | 1141 | | | | | | | | | | | | | | | | | | | | | | | | | | | [7] |
| 18 | F | 53 | AML | Allo | | | | 1643 | | | | | | | | | | | | | | | | | | | | | | | | | | | [7] |
| 19 | M | 50 | AML | Allo | | | | 4072 | | | | | | | | | | | | | | | | | | | | | | | | | | | [7] |
| 20 | F | 37 | CML | Allo | | | | 3173 | | | | | | | | | | | | | | | | | | | | | | | | | | | [7] |
| 21 | M | 39 | CML | Allo | | | | 4864 | | | | | | | | | | | | | | | | | | | | | | | | | | | [7] |
| 22 | M | 21 | ALL | Allo | | | | 3930 | | | | | | | | | | | | | | | | | | | | | | | | | | | [7] |

**Table 2.** *Cont.*

| Case Nr. | Sex | Age | Hem Dis | AI/Au | Diagnostic | | | Lab | Symptoms | | | | | | Treatment | | | | | | | | Other Organs | | | | | | | | | Treat Res | | | | Ref. |
|---|---|---|---|---|---|---|---|---|---|---|---|---|---|---|---|---|---|---|---|---|---|---|---|---|---|---|---|---|---|---|---|---|---|---|---|---|
| | | | | | Bx | Radio | EMG | CK | Weak | Pain | Joint | Swel | Fever | RoM | GC | CyA | Tacro | IG | Other | MM | Az | MA | Skin | Oral | Eyes | GI | Pulm | Liver | CNS | Joints | Heart | Res | IwD | NR | NCT | |
| 23 | M | 23 | AML | Allo | ■ | ■ | | 2882 | ■ | | | ■ | | ■ | | | | | | | | | ■ | | | | | ■ | | | | | ■ | | | [7] |
| 24 | M | 39 | ALL | Allo | ■ | | | 1790 | ■ | | ■ | | | ■ | | | | | ■ | | | | ■ | | | | | | | | | | ■ | | | [7] |
| 25 | F | 56 | AML | Allo | | | | NA | ■ | | | ■ | | | | ■ | | | | | | | | ■ | | | | | ■ | | | | | ■ | | | [7] |
| 26 | M | 57 | MM | Allo | | | | 25 | | | ■ | ■ | | | ■ | | | | | | | | ■ | | | | | ■ | | | | | ■ | | | [7] |
| 27 | M | 34 | AML | Allo | ■ | | | 69 | | | | ■ | | | | ■ | | | | | | | | | ■ | | | ■ | | | | | | ■ | | | [7] |
| 28 | F | 50 | AML | Allo | | ■ | | NA | | | ■ | | | | ■ | | | | | | | | ■ | | | | | | | | | | | ■ | | | [7] |
| 29 | F | 39 | MDS | Allo | ■ | | | 27 | | | | | | | | | | | | ■ | | | | | | | | | | | | | | ■ | | | [7] |
| 30 | F | 39 | CML | Allo | | ■ | | 171 | ■ | ■ | | ■ | | ■ | | | | | | | | | | | | | | ■ | | | | | ■ | | | [7] |
| 31 | F | 40 | AML | Allo | ■ | | | 64 | | | | ■ | | | ■ | | | | | | ■ | | ■ | | | | | ■ | | | | | ■ | | | [7] |
| 32 | M | 26 | ALL | Allo | ■ | | | 114 | ■ | | | | | | ■ | | | | | | ■ | | ■ | | | | | | | | | | ■ | | | [7] |
| 33 | M | 20 | CML | Allo | | ■ | | NA | | | | | | | ■ | | | ■ | | | ■ | | ■ | | | | | ■ | | | | ■ | | | | [7] |
| 34 | M | 49 | NHL | Allo | ■ | ■ | | 73 | ■ | | | | | ■ | | | | | | | | | | | | | | ■ | | | | ■ | | | | [8] |
| 35 | M | 4 | AA | Allo | ■ | | | 634 | | | | | | ■ | | | | | | | ■ | | ■ | | | | | ■ | | | | ■ | | | | [9] |
| 36 | F | 26 | CML | Allo | | | | 2350 | ■ | | | | | | | | | | | | | | ■ | | | | | ■ | | | | ■ | | | | [9] |
| 37 | F | 36 | CML | Allo | ■ | | | 54 | ■ | | | | | ■ | | | | | | | | | ■ | ■ | | ■ | | | | | | ■ | | | | [9] |
| 38 | M | 28 | CML | Allo | | | | NA | ■ | ■ | | | | | | | | | | | ■ | | ■ | | | | | ■ | | | | | | | | [9] |
| 39 | M | 46 | AML | Allo | ■ | | | 454 | ■ | | ■ | | | | | | | | | | | | ■ | | | | | ■ | | | | | | | | [9] |
| 40 | F | 19 | ALL | Allo | ■ | | | NA | ■ | ■ | | | | | | | | | | | ■ | | ■ | | | | | | | | | | | | | [9] |
| 41 | F | 8 | AA | Allo | ■ | | | 8400 | ■ | | | | | | | | | | | ■ | | | ■ | | | | | ■ | | ■ | | | | | | [9] |
| 42 | M | 36 | AA | Allo | | | | 2170 | ■ | ■ | | | | | ■ | | | | | | ■ | | ■ | | | | | ■ | | | | ■ | | | | [9] |
| 43 | M | 28 | CML | Allo | ■ | | | 49 | ■ | | | | | | ■ | | | | | | | | ■ | | | | | ■ | | | | ■ | | | | [9] |
| 44 | M | 24 | ALL | Allo | ■ | | | 2490 | ■ | | | | | ■ | | | | | | | ■ | | ■ | | | | | ■ | | | | ■ | | | | [9] |

**Table 2.** *Cont.*

| Case Nr. | Sex | Age | Hem Dis | AI/Au | Diagnostic | | | Lab | Symptoms | | | | | | Treatment | | | | | | | | Other Organs | | | | | | | | | Treat Res | | | | |
|---|---|---|---|---|---|---|---|---|---|---|---|---|---|---|---|---|---|---|---|---|---|---|---|---|---|---|---|---|---|---|---|---|---|---|---|---|
| | | | | | Bx | Radio | EMG | CK | Weak | Pain | Joint | Swel | Fever | RoM | GC | CyA | Tacro | IG | Other | MM | Az | MA | Skin | Oral | Eyes | GI | Pulm | Liver | CNS | Joints | Heart | Res | IwD | NR | NCT | Ref. |
| 45 | M | 42 | AML | Allo | | | | 1126 | ▨ | | | | | | ▨ | | | | | | | | ▨ | | | | | ▨ | | | | ▨ | | | | [9] |
| 46 | F | 35 | AML | Allo | ▨ | | | 81 | ▨ | | | | | | ▨ | | | | | | ▨ | | ▨ | | | | ▨ | | | | | ▨ | | | | [9] |
| 47 | F | 55 | AML | Allo | ▨ | | | NA | | | | | | | ▨ | | | | | ▨ | | ▨ | | | | | | | | | | ▨ | | | | [10] |
| 48 | M | 63 | AML | Allo | | | | 257 | | | | | | | ▨ | | ▨ | | | | | ▨ | | | | | | | | | | ▨ | | | | [10] |
| 49 | M | 66 | MDS | Allo | | | | 251 | | | | | | | ▨ | | ▨ | | | | | | | | | | | | | | | ▨ | | | | [10] |
| 50 | F | 54 | ALL | Allo | | | | 492 | | | | | | | ▨ | | | | | | ▨ | | | | | | | | | | | | | | ▨ | [10] |
| 51 | M | 45 | CLL | Allo | ▨ | | | 1853 | | | | | | | ▨ | | ▨ | | | | | | | | | | | | | | | ▨ | | | | [10] |
| 52 | M | 62 | AML | Allo | | | | 2195 | | | | | | | ▨ | | | | | ▨ | | | | | | | | | | | | ▨ | | | | [10] |
| 53 | M | 29 | AML | Allo | | ▨ | | N | | | | | | | ▨ | | | | | | | ▨ | | | | | | | | | | ▨ | | | | [11] |
| 54 | M | 39 | ALL | Allo | ▨ | | | 1220 | ▨ | | | | | | ▨ | | | ▨ | | | | | | | | | | | | | | ▨ | | | | [12] |
| 55 | F | 35 | AML | Allo | ▨ | | ▨ | 664 | | | | | | | ▨ | | | | | | | ▨ | | | | | | | | | | ▨ | | | | [13] |
| 56 | M | 47 | CML | Allo | ▨ | | ▨ | 2808 | ▨ | | | | | | ▨ | ▨ | | | | | | | | | | | | | | | | ▨ | | | | [13] |
| 57 | F | 48 | AML | Allo | ▨ | | | 2118 | | | | | | | ▨ | | ▨ | | | ▨ | | | | | | | | | | | | ▨ | | | | [14] |
| 58 | F | 31 | AML | Allo | ▨ | | | 18 | ▨ | | | | | | ▨ | | ▨ | | | | | | | | | | | | | | | | | ▨ | | [14] |
| 59 | M | 22 | AML | Allo | ▨ | | | 650 | | | | | ▨ | | ▨ | | | | | | ▨ | | | | | | | | | | | ▨ | | | | [15] |
| 60 | M | 51 | MM | Allo | ▨ | | | 946 | | | | | | | ▨ | | | ▨ | | | | | | | ▨ | | ▨ | | ▨ | | | ▨ | | | | [16] |
| 61 | M | 66 | AML | Allo | | | | N | | | | | | | ▨ | | | ▨ | | | ▨ | | | ▨ | | | | | | | | ▨ | | | | [17] |
| 62 | M | 63 | MDS | Allo | ▨ | | | N | ▨ | | | | | | ▨ | | | ▨ | | | | | | | | | | | | | | | | | ▨ | [17] |
| 63 | M | 33 | ALL | Allo | ▨ | | | 4× N | | | | | | | ▨ | | | | | | | | | | | | | | | | | ▨ | | | | [18] |
| 64 | F | 18 | AML | Allo | | ▨ | | 14420 | ▨ | | | ▨ | ▨ | | | | | ▨ | | | | | ▨ | | | | | | | | | | | | | | [19] |
| 65 | F | 37 | AML | Allo | ▨ | | | 1445 | | | | | | | ▨ | | | | | | | | | ▨ | | | | | | | | ▨ | | | | [20] |
| 66 | M | 54 | AML | Allo | ▨ | | | 358 | | | ▨ | | | | ▨ | | | | | | | | | | | | | | | | | ▨ | | | | [21] |
| 67 | M | 31 | CML | Allo | ▨ | | | NA | ▨ | | | | | | ▨ | | | | | | | | | ▨ | | | | | | | | ▨ | | | | [22] |

**Table 2.** *Cont.*

| Case Nr. | Sex | Age | Hem Dis | Al/Au | Diagnostic Bx | Radio | EMG | Lab CK | Weak | Pain | Joint | Swel | Fever | RoM | GC | CyA | Tacro | IG | Other | MM | Az | MA | Skin | Oral | Eyes | GI | Pulm | Liver | CNS | Joints | Heart | Res | IwD | NR | NCT | Ref. |
|---|---|---|---|---|---|---|---|---|---|---|---|---|---|---|---|---|---|---|---|---|---|---|---|---|---|---|---|---|---|---|---|---|---|---|---|---|
| 68 | M | 11 | ALL | Allo | ● | | ● | 3950 | ● | | | | | | ● | | | | | | | | | | | | | | | | | | | | ● | | [23] |
| 69 | M | 39 | AML | Allo | ● | | | 28 | | | | | | | ● | | ● | | | | | | | | | | | | | | | | ● | | | | [24] |
| 70 | F | 39 | CML | Allo | ● | | ● | 67 | | | | | | | ● | | ● | | | | ● | | | | | | | | | | | | ● | | | | [24] |
| 71 | M | 44 | CLL | Allo | ● | | | 437 | | | | | | | ● | | ● | | ● | | | | | | | | | | | | | | ● | | | | [24] |
| 72 | F | 65 | AML | Allo | ● | | | E | | | | | | | | | ● | | | | | | | | | | | | | | | | ● | | | | [25] |
| 73 | F | 59 | AML | Allo | ● | | | E | | ● | | | | | | | ● | | | | | | | | | | | | | | | | ● | | | | [25] |
| 74 | F | 67 | MDS | Allo | ● | | | E | | | | | | | | | ● | | | | | | | | | | | | | | | | ● | | | | [25] |
| 75 | M | 6 | THA | Allo | ● | ● | | 4570 | | | | | | | ● | | | | | | | ● | | | | | | | | | | | ● | | | ● | [26] |
| 76 | M | 57 | NHL | Allo | ● | | | N | | | | | | | ● | | | | | | | ● | | | | | | | | | | | ● | | | | [27] |
| 77 | M | 36 | MDS | Allo | ● | | | E | | | | ● | | | ● | | | | | | | | | | | | | | | | | | ● | | | | [28] |
| 78 | M | 52 | AML | Allo | ● | | ● | 923 | | | | | | | ● | | | | | | | | | | | | | | | | | | ● | | | | [29] |
| 79 | F | 54 | AML | Allo | ● | | | 1411 | | | | | | | | ● | | | | | | ● | | | | | | | | | | | ● | | | | [30] |
| 80 | M | 67 | AML | Allo | | | | N | ● | | | | | | | | | | ● | | | | | | | | | | | | | | ● | | | | [31] |
| 81 | M | 33 | CML | Allo | ● | | | 3113 | | | | | | | ● | | | | | | | ● | | | | | | | | | | | ● | | | | [32] |
| 82 | M | 35 | AML | Allo | ● | | | 3500 | | | | | | | ● | | | | | | | | | | | | | | | | | | ● | | | | [32] |
| 83 | M | 19 | CML | Allo | ● | | | 2943 | | | | | | | | | ● | | ● | | | | | | | | | | | | | | ● | | | | [32] |
| 84 | M | 62 | CML | Allo | | | | 1291 | | ● | | | | | | | | | | | | | | | | | | ● | | | | ● | | | | PRES |
| 85 | F | 13 | AA | Allo | ● | | ● | 1100 | | | | | | | | | | | | | | ● | | | | | | | | | | ● | | ● | | [33] |

This table demonstrates the demographical and clinical data for the 85 patients included in the study. Diagnostic describes the diagnostic work-up performed. Lab indicates CK values in U/L. Symptoms describe the symptomatology of the described patients. Treatment indicates the treatment given. Other organs describe the other organ systems known to be affected. Treat Res indicates the treatment outcome. The gray color indicates the positive result. Abbreviations: AML, Acute myelogenous leukemia; NHL, Non-Hodgkin's lymphoma; AA, Aplastic anemia; CLL, Chronic lymphocytic leukemia; MDS, Myelodysplastic syndrome; MM, Multiple myeloma; Thal, Thalassemia; Hem dis, Hematologic disease not specified Al/Au, Allogen/Autolog; Bx, Biopsy; MRI, Magnetic resonance imaging; CT, Computed Tomography Scan; EMG, Electromyography; Weak, Muscle weakness; Pain, Muscle pain; Joint, Joint pain; Swel, Muscle swelling; RoM, Range of motion; GC, Glucocorticoids; CyA, Cyclosporin A; Tacro, Tacrolimus; IG, Intravenous immunoglobulin; MM, Mycophenolate mofetil; Az, Azathioprine; MA, Monoclonal antibodies; Res, Resolved; IwD, Improvement with disability; NR, Not responsive; NCT, Not completed treatment; HUS, Haukeland University Hospital; N, normal; N/A, Not Available; E, Elevated; F. Female; M. Male; and Treat Res, Treatment result.

**Table 3.** This table shows the distribution of patients regarding gender and hematological diseases.

|  | Number (*n*) | Percentage (%) |
|---|---|---|
| **Sex** | | |
| Male | 53 | 62 |
| Female | 32 | 38 |
| **Hematological Diseases** | | |
| Acute myelogenous leukemia (AML) | 33 | 39 |
| Chronic myelogenous leukemia (CML) | 15 | 18 |
| Acute lymphocytic leukemia (ALL) | 11 | 13 |
| Myelodysplastic syndrome (MDS) | 7 | 8 |
| Aplastic anemia (AA) | 5 | 6 |
| Non-Hodgkin lymphoma (NHL) | 5 | 6 |
| Chronic lymphocytic lymphoma (CLL) | 4 | 5 |
| Multiple myeloma (MM) | 3 | 4 |
| Mantle cell lymphoma (ML) | 1 | 1 |
| Thalassemia (Thal) | 1 | 1 |

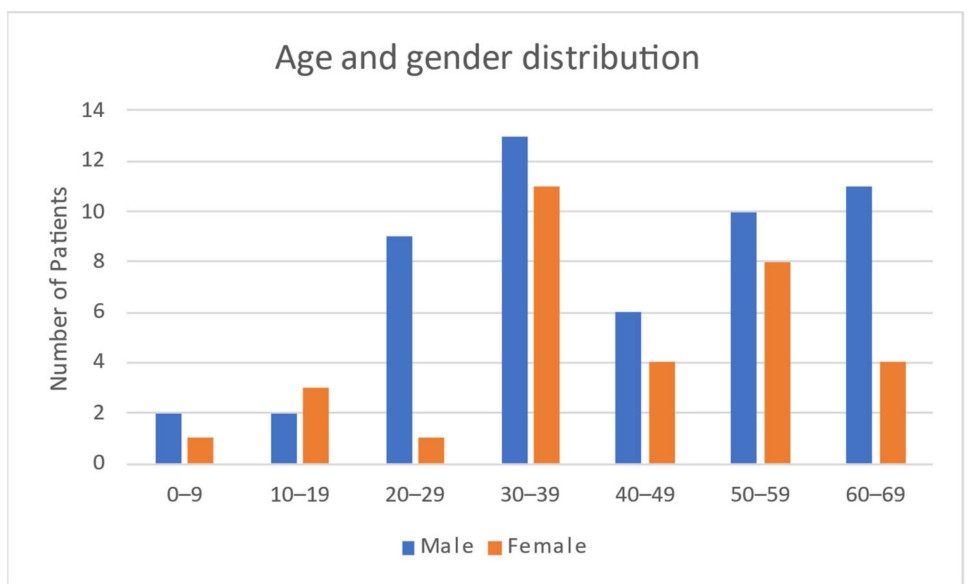

**Figure 2.** This figure represents gender and age distribution among the described cases.

### 4.2. Symptoms and Clinical Features

We found that musculoskeletal cGVHD could present with various symptoms. The most common were muscle weakness (81% of the patients) and muscle pain (60%). Other symptoms were fever, swelling or edema, restriction of movement, and muscle tenderness.

Investigations performed for the confirmation of musculoskeletal cGVHD were mainly biopsy, blood tests, EMG, and various imaging modalities. Blood tests were the most common, with all but one (99%) having one or more tests performed. A biopsy was the second most common, with 76 (89%) of the patients having one or more muscle biopsies performed. Imaging included MRI, CT, and X-ray: 30 (35%) of the 83 patients had imaging as a part of the diagnostic workup; in 27, this was a MRI, two had a CT scan, and the last had an X-ray. EMG was performed on twenty-two (26%) patients. Only one (1%) patient had a musculoskeletal cGVHD diagnosis made on a clinical basis. We found that one (1%) article did not enclose the diagnostic workup and was therefore classified as unknown. The majority of patients (18, 21%) had the CK measured (Table 2); however, the variations in the serum levels of CK demonstrated great distribution among the patients described, with a median 1570, ranging from 5 to 14,420 (Figure 3). It is also important to note that 27 (32%)

of patients presented with one or more musculoskeletal symptoms, with normal values of CK.

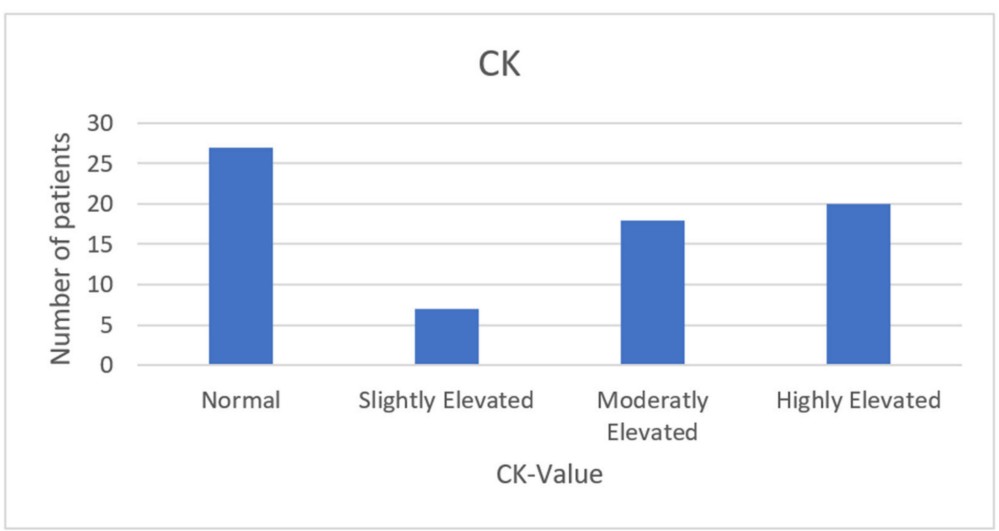

**Figure 3.** The table shows the distribution of CK values at the time of diagnosis of chronic musculoskeletal GVHD. The following cutoff values were used: Normal, <280 U/L; slightly increased, 280–500 U/L; moderately increased, 500–2000 U/L; and strongly increased, >2000 U/L.

*4.3. Affection of Other Organs*

Data about cGVHD simultaneously affecting other organs were also collected in this review. The data are shown in Figure 4. The majority of the 85 patients showed signs of GVHD in other organs outside of the musculoskeletal system. Most (61, 72%) of the 85 individuals had diseases in at least one other system. From these 61 patients, 18 (21%) had disease in a single organ, while 43 (51%) had disease in multiple organs. Skin was the most common organ affected, with 44 (52%) individuals affected. Oral and pulmonary followed as the second and third.

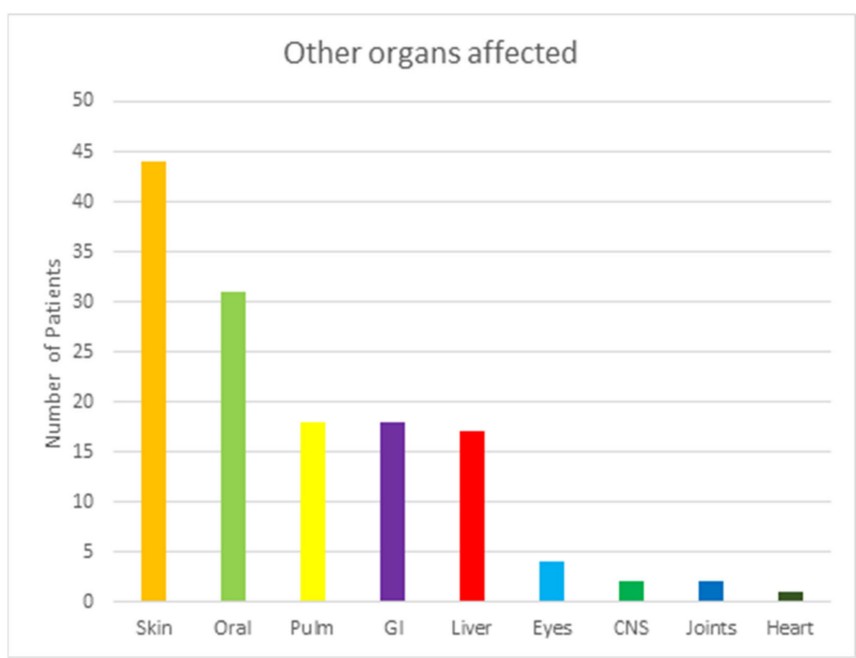

**Figure 4.** This table shows other organs affected by GVHD among patient with musculoskeletal GVHD. Abbreviations: Pulm, Pulmonary; GI, Gastrointestinal tractus; and CNS; Central Nervous System.

### 4.4. Treatment

The main treatment for musculoskeletal GVHD is corticosteroids, either as monotherapy or in combination with other interventions (Figure 5). During this systematic review, we found a total of 39 different treatment approaches based on the combination of medications. Four patients had insufficient data concerning the medications used and were therefore classified as unknown. To simplify the available data, all medications were divided into groups based on their class of drug. The usage of the different drugs is listed in Figure 4. Combination treatment was the most common intervention, with 65 (77%) of patients having two or more medications. Prednisolone and tacrolimus were most frequently used of all the combinations, with 12 (14%) of all patients having this therapy. Following that was prednisolone and cyclosporin in nine patients (11%) and prednisolone and azathioprine in six patients (7%). Prednisolone was the most frequent monotherapy, with 10 patients (12%) using this treatment.

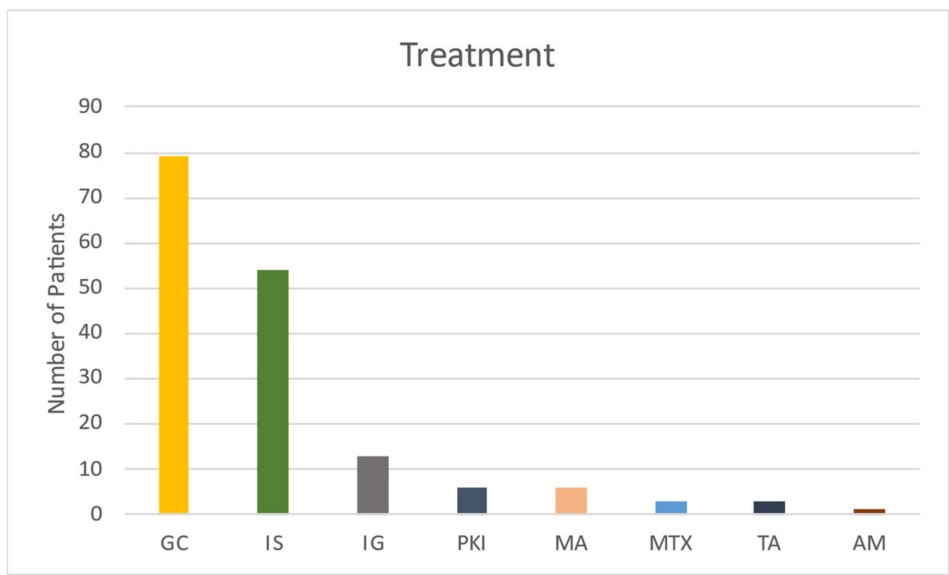

**Figure 5.** The table shows the distribution of medical treatment done in patients with chronic musculoskeletal GVHD. Abbreviation: GC, Glucocorticoids; IS, Immunosuppressant; IG, Immunoglobulins; PKI, Protein Kinase Inhibitor; MA, Monoclonal Antibodies; MTX, Methotrexate; TA, Therapeutic Apharesis; AM, Antimalarial. Designation: Glucocorticoids, Prednisolon, Methylprednisolon, and Budenoside; Immunosuppressant, Ciclosporin, Tacrolimus, Azathioprine, Mycophenolate Mofetil, Leflunimid, and Cyclophosphamide; Immunoglobulins, Polyvalent Immunoglobulins, and Gamma globulin; Protein Kinase Inhibitor, Ruxolitinib, and Ibrutinib; Monoclonal Antibodies, and Rituximab; MTX, Methotrexate; and Therapeutic apharesis, Plasmaexchange, and Extracorpeoral Photopheresis.

### 4.5. Outcome and Survivability

The outcome following treatment appeared mostly positive.Complete resolution occurred in 49 (57%), while 19 (22%) showed improvement together with some degree of disability. Only one (1%) patient failed to respond, while another did not complete treatment due to financial reasons. Fifteen patients (fourteen of them from the same article) had insufficient information regarding their treatment response and were therefore classified as unknown in this review.

Of the 85 patients, 64 were alive at the time of writing each respective article this review is based on. The timeframe given from the various articles ranged from 3 months after the GVHD diagnosis was made to 25 years. The remaining 21 patients were deceased at the time of writing the source articles. Of these 21 patients, four completely recovered after GVHD treatment. Five patients had improvements in various disabilities, one patient was unresponsive to treatment, and one did not complete treatment due to financial reasons.

The remaining 10 did not have recordings of a treatment response written in the source article. The times of death ranged from one month after allogeneic stem-cell transplant to 6.5 years. Only one of these 21 patients had a cause of death recorded (bacterial sepsis). Therefore, the remaining deaths could not be directly linked to the allo-HSCT or the musculoskeletal cGVHD diagnosis.

## 5. Discussion

Musculoskeletal cGVHD is a well-known yet rare manifestation following allo-HSCT, with an incidence ranging from 0.5% to 3% [3,7]. Thus, the number of available clinical studies were few.

While our understanding of the pathophysiology of aGVHD has grown substantially, the mechanisms involved in cGVHD remain poorly understood. Greater understanding of these mechanisms has also been hampered by the lack of animal models that accurately mimic the disease [34]. Nevertheless, several hypotheses based on experimental studies have been presented. These include T-regulatory cell deficiency, defects in T-cell selection due to thymus damage, anomalous B-cell behavior, leading to autoantibody production and profibrotic lesions [2,34]. There appears to be increasing evidence that damage to the thymus is a leading cause of cGVHD after an allo-HSCT. Following allo-HSCT, T-cell recovery in the recipient is a result of two pathways: one depends on the thymus, while the other is thymus-independent [5]. The pathway that relies on the thymus affects the creation of naïve T cells, from stem cells that originated from the donor. Thus, any damage to the thymus due either to the treatment, disease, or age-related atrophy can hamper the development of functioning T cells. Prior aGVHD has been shown to greatly increase the risk of cGVHD, primarily by the damage of medullary thymic epithelial cells [35]. It is also worth noting that cGVHD usually does not transpire following an auto-HSCT, where thymopoiesis is regularly observed [5,36,37].

The complications of the disease can severely affect function and quality of life and therefore are an important cause of increased morbidity following allo-HSCT [7]. Interestingly, cGVHD has also been shown to be protective in regards to a relapse of the underlying malignant disease. Therefore, the overall survival rate of patients is reflected in the balance of its negative, i.e., treatment-related morbidity and mortality and positive, i.e., lower incidence of relapse [38]. In order to diagnose cGVHD, a biopsy that reads "consistent with" GVHD paired with at least one distinctive clinical feature is required [2,34,39,40]. Our patient presented with the clinical symptoms of muscle pain and stiffness and a confirmed histopathology. In most cases reviewed, a biopsy was performed to verify the diagnosis, which is recommended by international guidelines. However, cGVHD presenting as fasciitis can lead to restriction of movement, either primarily or secondary to deep sclerosis [7]. This, according to the national institute of health (NIH)s criteria from 2005, is enough to determine the diagnosis of cGVHD. Myositis or polymyositis, showing symptoms of muscle weakness, muscle pain, and/or edema, may also be the indicator of cGVHD (Table 4). These patients require additional biopsy or radiology examination to conclude the correct diagnosis. It is also important to rule out any drug-induced myopathy, such as iatrogenic glucocorticoid-induced myopathy. This is the most common drug-induced myopathy and may overlap with other symptoms of musculoskeletal cGVHD [41]. The risk factors associated with cGVHD are related to the history of aGVHD, the donor, and the use of total body irritation [38]. As stated above, a previous history with aGVHD and its severity is the main predictor. Donor risk factors are linked to mismatched donors, older donor ages, and peripheral blood stem cells (PBSCs) as donor sources. Besides cGVHD-damaging effects, it has also been shown to be protective regarding the recurrence of the underlaying malignancy [42]. According to Grube et al., the increased severity of cGVHD is associated with a lower incidence of relapse in cases without the GI tract or pulmonary or liver involvement [42].

Table 4. This table shows part of the NIH table suggesting new criteria for diagnosing cGVHD.

| Organ or Site | Diagnostic (Sufficient to Establish the Diagnosis of cGVHD) | Distinctive (Seen in cGVHD, but Insufficient Alone to Establish a Diagnosis of cGVHD) | Other Features * |
|---|---|---|---|
| Muscles, fascia, joints | Fasciitis, joint stiffness, or contractures secondary to sclerosis | Myositis or polymyositis ** | Edema, muscle cramps, arthralgia, or arthritis |

* Symptoms that can be acknowledge as part of cGVHD if a diagnosis is made. ** Clinical manifestations that require biopsy or radiology to confirm the diagnosis. Abbreviations: cGVHD, chronic graft versus host disease.

In this article, we presented a patient who developed liver cGVHD followed by musculoskeletal affection. Musculoskeletal manifestation comes in a variety of forms, such as myositis, fasciitis, joint stiffness, and contractures [4]. The clinical presentation of musculoskeletal cGVHD varies, but the symptoms range from muscle weakness, muscle pain, fever, swelling in the affected muscles, edema, reduced joint range of motion, and muscle tenderness. Usually, in the early inflammatory stages of cGVHD, the patient will present with muscle tenderness and edema, which, if left untreated, progresses to contractures and fibrosis [38]. In our literature review, we found the most frequent symptoms to be muscle weakness and muscle pain. Our patient had muscle pain and stiffness affecting the proximal muscles. Additionally, we found that a significant portion of the patients with musculoskeletal cGVHD had a coexistent involvement of other organs. Most (62, 73%) of the 85 individuals reviewed had cGVHD manifestation in at least one other organ system. The most frequent other organs affected were skin with four (51%), followed by the oral cavity, lungs, GI tract, and liver. These findings are also in agreement with other findings of frequent organ systems affected by cGVHD.

Even though musculoskeletal involvement is rare, it is important to consider the diagnosis in patients with appropriated clinical features, as the condition is potentially life-threatening. Furthermore, initiating treatment early is essential to prevent irreversible fibrosis. Out of 85 patients, 21 were deceased, with only one having a cause of death, bacterial sepsis, recorded. It is not possible, therefore, to link the remaining deaths to musculoskeletal cGVHD. While the prognosis varies greatly depending on the associated underlying disease, a substantial number of patients (49/85, 58%) achieved complete remission after being treated for musculoskeletal cGVHD, while 19 (22%) patients showed an incomplete recovery, and 1/85 (1%) had no treatment response. The development of profibrotic lesions and contractures is a poor prognostic event, and the reason why it is important to consider early physiotherapy treatment, including myofascial massage and stretching, in order to restore and maintain a range of motion in the affected joints and muscles [42,43]. To our knowledge, no study has evaluated physiotherapy as part of a treatment regimen; hence, the effects of this remain uncertain.

No diagnostic or prognostic biomarker for cGVHD in general or musculoskeletal cGVHD in special exist at the present time. CK is frequently measured in muscle disease, although only 32% of patients with musculoskeletal cGVHD presented with normal values of CK. Thus, CK is a poor diagnostic marker but can still be used as an indicator of the diagnosis. Measurement of the myoglobin or myoglobinuria could be another approach, although, in very few cases, this was measured. General markers of inflammation such as CRP and ferritin could be used, although would in general reflect unspecified inflammatory conditions [44,45]. New biomarkers are entering the research fields of cGVHD [46,47], although if they have a diagnostic or prognostic impact in musculoskeletal cGVHD remains unanswered.

Although the first-line treatment for cGVHD is corticosteroids, polytherapy appeared to be the most common therapeutic action, with 60 (71%) of the patients receiving more than one drug. The most frequent medication combinations were prednisolone paired with tacrolimus (12, 14%) and prednisolone with cyclosporin (9, 11%). This is in accordance with

the general treatment recommendation for cGVHD. Therefore, it is quite interesting how well our patient responded to the monotherapy with ruxolitinib. Corticosteroids can, after all, cause severe toxicity over time. Finally, it should also be emphasized how important an interdisciplinary team consisting of hematologists, radiologists, neurophysiologist, neurologists, and physiotherapists is for the treatment and diagnostic workup of musculoskeletal cGVHD [2,3,43].

## 6. Conclusions

Even though musculoskeletal cGVHD is rare and the existing literature scarce, the disease has distinct clinical signs and symptoms, and the development of myositis or fasciitis following allo-HSCT should lead to a diagnostic workup to rule out the possibility of musculoskeletal cGVHD. The discovery of this at an early stage is important to avoid irreversible damage. Furthermore, given the vast potential for complications in GVHD, an interdisciplinary team might be helpful in preventing severe consequences of cGVHD and improve the patient's prognosis [43].

**Supplementary Materials:** The following supporting information can be downloaded at: https://www.mdpi.com/article/10.3390/curroncol29110663/s1: Figure S1: Flowchart demonstrating the selection of articles used for the systemic review.

**Author Contributions:** Conceptualization, A.D.K., T.K. and H.R.; methodology A.D.K., T.K. and H.R.; software A.D.K., T.K. and H.R.; validation, A.D.K., T.K. and H.R.; formal analysis, A.D.K., T.K. and H.R.; investigation, A.D.K., T.K., H.M., L.A.B. and H.R.; resources, H.R.; data curation, A.D.K., T.K. and H.R.; writing—original draft preparation, A.D.K., T.K. and H.R.; writing—review and editing, A.D.K., T.K., H.M., L.A.B. and H.R.; visualization, A.D.K., T.K., H.M. and H.R.; supervision, H.R.; project administration, H.R.; and funding acquisition, H.R. All authors have read and agreed to the published version of the manuscript.

**Funding:** This research received no external funding.

**Informed Consent Statement:** Informed consent was obtained from the patient presented as part of the study.

**Acknowledgments:** We are grateful for the patients and their next of kins' willingness to publish this article.

**Conflicts of Interest:** The authors declare no conflict of interest. The funders had no role in the design of the study; in the collection, analyses, or interpretation of the data; in the writing of the manuscript; or in the decision to publish the results.

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
