# Peer review of "Musculoskeletal Chronic Graft versus Host Disease—A Rare Complication to Allogeneic Hematopoietic Stem Cell Transplant: A Case-Based Report and Review of the Literature"

_curroncol, doi:10.3390/curroncol29110663_

Round 1

Reviewer 1 Report

1.Myoglobin is an important indicator of muscle myopathy.What was the myoglobin level in this case?

2. In the case,the patient continues to exhibit abnormally high 

ferritin following treatment. Is there a reason for that? Are the  muscular symptoms in this patient an advanced demonstration of a malignant tumour?

3.In Figure 4, what is the specific value of the X-axis?

4.Is there an association between ck values and musculoskeletal GVHD?

Author Response

1.Myoglobin is an important indicator of muscle myopathy. What was the myoglobin level in this case?

Regarding myoglobin, we agree that this could be of interest in the present cases of muscle GVDH, however we gain review the literature and only three patients have measured myoglobin and two have measured myoglobinuria. Hence, we have we left this out of the present description we have shortly discussed myoglobin as measurement for muscle GVHD in our revised version of the paper.

  1. In the case,the patient continues to exhibit abnormally high ferritin following treatment. Is there a reason for that? Are the  muscular symptoms in this patient an advanced demonstration of a malignant tumour?

It is correctly observed. Elevated ferritin could have several reasons in the allo-HSCT setting, such as transfusion dependent iron overload, liver toxicity, general inflammation, aGVHD, and cGVHD. We believe it is non-sensitive marker, we have discussed these features shortly in the discission section.

3.In Figure 4, what is the specific value of the X-axis?

We agree that the figure was somehow unprecise and difficult to interpret. In the revised version of the manuscript, we have altered the description to normal (<280 U/L), slightly increased (280-500 U/L), moderately increased (e.g. 500-2000 U/L) and strongly increased (>2000 U/L). We think this presentation is more suitable.

4.Is there an association between ck values and musculoskeletal GVHD?

Since 32% of patients presented with one or more musculoskeletal symptoms, with normal values of CK, we believe CK is a poor diagnostic marker musculoskeletal GVHD, and we have highlighted this in the manuscript.

Reviewer 2 Report

Kvinge et al describes a case report and then describe the literature available and published about Musculoskeletal GVHD. 

1. Needs some english language corrections e.g. tractus (pg 1-line 37, page 2-line 55)

2. A number of details are missing from the case such as overall grade of acute GVHD- GIT stage, skin stage etc. Also any biopsy was obtained.

3. The case also requires additional details, regarding treatments with dose of prednisone and Ruxolitinib. Details about day of onset post transplant and day of resolution of GVHD or day of treatment instituted.

4. What were the symptoms of presentation of musculoskeletal GVHD? What differential diagnosis was thought of and excluded- some details may be helpful to the reader.

5. Donor characteristics and HLA matching info not available. What muscles were involved in GVHD.

6. EMG done? Chimerism status? outcomes of treatment with steroids, Rux?

7. Fig 2 is not required. And also selection criteria is not required in detail.

8. Shorten table 2 and summarize the symptoms and group the cases. This table may take up a lot of space.

9. Fig 3, 4 and 7 are not required and the data can be mentioned in manuscript. 

10. Similarly, the paragraph from line 246 till 255 can be shortened.

This is a well thought review on musculoskeletal GVHD but requires major revision to be considered for publication. 

Author Response

Musculoskeletal GVHD. 

  1. Needs some english language corrections e.g. tractus (pg 1-line 37, page 2-line 55)

Minor spell check is performed, and typos corrected.

  1. A number of details are missing from the case such as overall grade of acute GVHD- GIT stage, skin stage etc. Also any biopsy was obtained.

We agree that there was a somewhat limited description of some details in the medical history, this has now been improved in our revised version of the manuscript

  1. The case also requires additional details, regarding treatments with dose of prednisone and Ruxolitinib. Details about day of onset post transplant and day of resolution of GVHD or day of treatment instituted.

More details are added.

  1. What were the symptoms of presentation of musculoskeletal GVHD? What differential diagnosis was thought of and excluded- some details may be helpful to the reader.

We symptom burden of our patients is described in more details, as differential diagnosis considered in the present case are discussed.

  1. Donor characteristics and HLA matching info not available. What muscles were involved in GVHD.

We have added characteristics regarding donor situations and chimerism status, and aspects of muscle biopsy.

  1. EMG done? Chimerism status? outcomes of treatment with steroids, Rux?

EMG was don’t performed. Treatment outcome is described.

  1. Fig 2 is not required. And also selection criteria is not required in detail.

Selection criteria is revised and shortened. Former Figure 2 is moved to Supplementary.

  1. Shorten table 2 and summarize the symptoms and group the cases. This table may take up a lot of space.

We agree that Table 2 contained much data, and we have simplified this a bit in our revised version. However, we will argue that is an important table given a precisely overview of the data. We will therefore argue to keep Table 2 in this manuscript.

  1. Fig 3, 4 and 7 are not required and the data can be mentioned in manuscript. 

Previously Figure 3 is simplified, and previously Figure 7 is removed. Previously Figure 4 however we believe is important for the manuscript and the visualization for the data, and we therefore have kept this Figure in our revied version of the manus.

  1. Similarly, the paragraph from line 246 till 255 can be shortened.

We agree in this comment, and accordingly have left this paragraphs out.

This is a well thought review on musculoskeletal GVHD but requires major revision to be considered for publication. 

Reviewer 3 Report

Muskuloskeltal cGVHD is a rare complication of allogeneic stem cell transplantation so that awareness regarding such infrequent manifestations is very important. 

Any article that provides information on rare and lesser known manifestations is an important source of information.

In addition, the authors conducted an extensive review of the data from the literature related to musculoskeletal GVHD and presented extensively the heterogeneous clinical picture, the appearance at variable time intervals after transplantation, the diagnostic methods and the various therapies used in the absence of a standardized therapeutic approach.

I recommend publishing the article in its current form.

Author Response

Muskuloskeltal cGVHD is a rare complication of allogeneic stem cell transplantation so that awareness regarding such infrequent manifestations is very important. 

Any article that provides information on rare and lesser known manifestations is an important source of information.

In addition, the authors conducted an extensive review of the data from the literature related to musculoskeletal GVHD and presented extensively the heterogeneous clinical picture, the appearance at variable time intervals after transplantation, the diagnostic methods and the various therapies used in the absence of a standardized therapeutic approach.

I recommend publishing the article in its current form.

We are grateful for the positive comments regarding our manuscript. Minor alterations are performed based on the other reviewers’ reports.